# Codify and Localize Lesions on a Coronary Acoustic Map: Scientific Rationale, Trial Design and Artificial Intelligence Algorithm Protocols

**DOI:** 10.3390/diagnostics15232994

**Published:** 2025-11-25

**Authors:** Thach Nguyen, Khiem Ngo, Hoang Anh Tien, Dzung T. Ho, Chinh D. Nguyen, Loc T. Vu, Mihas Kodenchery, Huynh Hung, Vinh X. Huynh, Aravinda Nanjundappa, Michael Gibson

**Affiliations:** 1Interventional Cardiology, Methodist Hospital, Merrillville, IN 46410, USA; drmihasmamu@yahoo.com (M.K.); hhlinda@yahoo.com (H.H.); 2St Mary Medical Center, Hobart, IN 46342, USA; 3School of Medicine, Tan Tao University, Tay Ninh 840000, Vietnam; triloc27@gmail.com (L.T.V.); xuanhuyhuynh2003@gmail.com (V.X.H.); 4Department of Geriatrics, University of Texas A&M, Temple, TX 78712, USA; luoikiemvang@gmail.com; 5Cardiovascular Centre, Hue University of Medicine and Pharmacy, Hue University, Hue 53000, Vietnam; bsanhtien@gmail.com; 6Interventional Cardiology, VinMec International Hospital, Ho Chi Minh City 700000, Vietnam; dunghothuong@yahoo.com; 7Interventional Cardiology, SIS International Hospital, Can Tho 94118, Vietnam; chinh0208@gmail.com; 8Interventional Cardiology, Cleveland Clinics, Cleveland, OH 44195, USA; nanjuna@ccf.org; 9Baim Institute of Clinical Research, Harvard Medical School, Harvard University, Boston, MA 02115, USA

**Keywords:** fluid mechanics, retrograde pressure wave, water hammer, recirculating flow, coronary acoustic map

## Abstract

In coronary artery disease (CAD), the initiation, progression, and regression of atherosclerosis remain incompletely understood, limiting the effectiveness of specific diagnostic and personalized medicine management strategies based on current imaging and assessment methods. In this scientific rationale and study design analysis, the framework conceptualizes the cardiovascular system as an integrated hydraulic network of pumps and pipes, advancing a shift from static imaging of luminal stenosis toward dynamic assessment of coronary flow. Grounded in fluid mechanics and acoustic principles, this analysis establishes a scientific rationale for an angiographic investigation of hemodynamic disturbances that compromise endothelial integrity in coronary arteries. The first section examines injury arising from repetitive flexion and extension of coronary segments driven by left ventricular contraction, most prominent at the transition from diastole to systole. The second section evaluates the hypothetical effects of thickened boundary layers and intimal injury caused by oxygen deprivation along the proximal portion of the outer curvature of side branches. The third section explores the hypothetical role of recirculating flow in accelerating lesion development at these sites. The fourth section presents an acoustic-based diagnostic framework for assessing the hypothetical impact of retrograde pressure-wave propagation associated with water-hammer phenomena. Collectively, these mechanisms establish the systematic codification and spatial delineation of coronary lesions as represented on the coronary acoustic map. Building on these insights, the present analysis proposes a clinical trial framework integrating AI-driven algorithmic protocols to rigorously assess the diagnostic performance and predictive accuracy of the coronary acoustic map.

## 1. Introduction

Over the past 75 years, the global shift from traditional home-prepared meals to diets dominated by cholesterol-dense, processed foods, together with the increasing prevalence of uncontrolled hypertension, has fueled the worldwide epidemic of coronary artery disease (CAD) [1]. The earliest step in atherosclerosis occurs with injury to the intima which lines the inner surface of the coronary arterial lumen. Once this protective endothelial barrier is compromised, low-density lipoprotein (LDL) cholesterol particles could penetrate the intima and accumulate in the subintimal space. With time, these lipid deposits aggregate and evolve into atherosclerotic plaques [2].

Since molecular biology has not established the mechanism of the above initial intimal injury, many researchers have adopted a different perspective—examining atherosclerosis through the principles of fluid mechanics (FM) [3]. The hemodynamic hypothesis of atherosclerosis was first articulated by Meyer Texon, M.D., in 1957 [4]. As a pathologist examining arterial lesions at specific anatomical sites, Texon proposed that atherosclerosis originates from localized fluid mechanical stresses acting on discrete arterial segments identified in cadaveric specimens. More than a decade later, in 1969, C. G. Caro reinforced and expanded this concept in a landmark *Nature* publication, asserting that regions of low wall shear stress (WSS) are critical in the initiation and progression of atherosclerotic plaques along the coronary arterial wall [5]. Although WSS has since become widely acknowledged as a principal hemodynamic determinant of atherogenesis, it remains unmeasured in modern cardiac catheterization laboratories.

The development of fractional flow reserve (FFR) and intracoronary Doppler techniques has provided indirect means of evaluating coronary physiology. FFR quantifies the pressure gradient across a lesion, while Doppler analysis estimates local flow velocity and direction at isolated single points along the coronary artery [6,7]. Yet, these methods fall short of capturing the integrated dynamics of coronary flow. Similarly, intravascular ultrasound (IVUS) and optical coherence tomography (OCT) deliver high-resolution structural delineation across the coronary arterial wall thickness but remain inherently static modalities, incapable of visualizing or quantifying intraluminal flow motion [8,9]. Coronary computed tomography angiography (CTA) offers volumetric assessment and flow velocity estimation; however, it cannot determine flow direction—such as retrograde propagation—or characterize flow configuration, such as laminar versus turbulent motion. This limitation stems from the static nature of CTA, which produces high-resolution three-dimensional reconstructions of coronary anatomy but lacks the temporal resolution necessary to capture real-time hemodynamic phenomena [10].

As a result, our team refined the existing coronary angiographic recording and reviewing techniques, enabling researchers to quantify and evaluate the impact of both normal and abnormal coronary flow dynamics [11]. This advancement, grounded in hydrodynamic principles, provided a mechanistic approach to delineate regions prone to disturbed flow, thereby integrating flow patterns, shear stress, and vascular pathology into a cohesive analytical framework.

In this scientific rationale and study design analysis, the format is structured to integrate cardiovascular pathology with hydrodynamic mechanisms. First, four mechanistic pathways derived from FM and acoustics are outlined as potential initiators of intimal injury. Second, fundamental flow phenomena in hydraulic systems are examined, emphasizing how they compromise the integrity of pumps, pipes or fluid channels leading to structural damages and flow obstruction. Third, FM and acoustic principles are analyzed within the context of coronary arteries to clarify how endothelial disruption may drive plaque initiation, progression, regression, and rupture.

Our objectives are twofold: (1) to apply FM and acoustic principles as analytical frameworks for elucidating the hypothetical mechanisms that drive the formation and progression of coronary lesions as presented on a coronary acoustic map, and (2) to use the resulting mechanistic insights as the basis for designing trials aimed at validating the hypothetical concepts on a broader scale.

## 2. Investigations for a Scientific Basis

### 2.1. Classifications

In general, our preliminary findings indicated that most coronary lesions cluster in four locations (#1 to #4) highlighted on the coronary acoustic map (Figure 1). The initiation and progression of these lesions were tentatively linked to four mechanistic pathways derived from FM and acoustic principles, which will be discussed in sequence based on ascending order of complexity (Table 1).

The first mechanistic pathway involves cyclic flexion of arterial segments during left ventricular (LV) contraction and relaxation. This repetitive bending concentrates mechanical stress within the intima, generating focal regions vulnerable to subintimal infiltration and deposition of LDL particles [13]. The second mechanism arises in regions of low shear stress along a curved arterial wall, where thick stagnant boundary layers impede oxygen replenishment and thereby reduce oxygen availability to the intima. This prolonged suboptimal hypoxic state may accelerate endothelial senescence and apoptosis, fostering calcification in the absence of significant luminal narrowing [14].

The third mechanism emerges at arterial bifurcations, where recirculating flow develops along the exit slope of the ostium of a secondary branch. Such disturbed flow fields intensify local shear stress gradients, precipitating recirculating flow, starting local vortices, injuring the local endothelial layers [15]. The fourth and most complex mechanism involves recurrent surges and drops in pressure generated by collisions between diastolic antegrade flow and systolic retrograde pressure waves. Do these water-hammer-driven oscillations impose alternating shear and pressure loads on the intima, initiating plaque formation, accelerating its progression, and increasing its susceptibility to rupture? In each case, angiographic images are presented first, followed by an investigation of the corresponding phenomena in hydraulic pumps and pipes through FM or acoustic analysis. The discussion concludes with application of the same analytic framework to coronary arteries.

### 2.2. Mechanical Stress Due to Repetitive Bending at a Hinge Location

Because the coronary artery follows the curvature of the LV, the trajectory of the artery may subject itself to hinge-like bending at specific sites.


**Mechanics Perspective:**


The progressive degradation of a hinge subjected to excessive use is primarily mechanical. Each actuation introduces friction between the hinge components, leading to surface abrasion and gradual material loss. Repeated cycles of loading and unloading impose stress on the metal elements, promoting fatigue and initiation of microscopic cracks that may propagate to structural failure. Moreover, recurrent movement can loosen screws or pins, increasing mechanical play within the assembly and thereby accelerating wear and predisposing the hinge to premature failure [16].


**Hinge Movement in Coronary Arteries:**


The repetitive localized bends intensify mechanical stress and strain concentration on the arterial wall while disturbing flow stability. The combination of structural stress and altered hemodynamics makes these sites vulnerable to injury, providing initiation points for atherosclerotic plaque development and subsequent rapid growth (Figure 2A,B and Appendix A).

This mechanism is further amplified in patients with anatomical hinge points located at the juncture of diastole transitioning to systole. At these sites, severe turbulence arises from superimposition of collision by the diastolic antegrade flow and systolic retrograde pressure waves generated from a water-hammer phenomenon. In the setting of ST-segment elevation myocardial infarction (STEMI), compensatory hyperdynamic contraction of the LV wall supplied by the non-infarct-related artery (IRA) intensifies mechanical stress at these hinge regions, often producing angiographic haziness that may be misinterpreted as a vulnerable lesion. Following reperfusion of the IRA, and particularly when LV contractility is attenuated by beta-blockade, the apparent severity of these hinge lesions frequently diminishes or resolves.

In real life situation, this understanding underscores the importance of cautious interpretation of angiographic findings in an acute STEMI setting, where transient hinge-related angiographic haziness may not represent true luminal stenosis. In a case report of a patient with inferior wall myocardial infarction, angiography before reperfusion suggested a severe lesion in the mid left anterior descending (LAD) artery; however, repeat angiography three months later demonstrated complete resolution of the haziness at the mid-LAD hinge point, confirming the dynamic rather than fixed nature of the lesions at hinge point. These transient changes are more likely to appear if the hinge are situated at location where diastole transitions to systole (Figure 3A–D and Appendix A).

### 2.3. Local Intimal Ischemia Due to Thick Boundary Layer


**Fluid Mechanics Perspective:**


In a straight channel or river stretch, the velocity field of the fluid stratifies into a high-velocity central layers (inner stream) with reduced static pressure and a low-velocity peripheral layers (outer stream) with elevated pressure, a distribution governed by wall shear and viscous boundary effects [17]. At a bifurcation, the hydrodynamic regime undergoes a marked reorganization. The main channel, which preserves a near-linear trajectory with minimal angular deviation, maintains flow attachment along the inner wall of the flow divider [17]. Under these conditions, the thresholds for large-scale flow detachment are not reached, and no macroscopic recirculation zone develops (flow at main channel in Figure 4).

In contrast, the fluid entering a distributary encounters a sharper angular deflection. By inertia, the central layers (inner stream) maintains their high-velocity, low-pressure trajectory without exerting erosive forces on the inner surface of the flow divider. Along the outer curve, however, velocity decay induced by wall friction generates a localized zone of high pressure and slow flow, seen as a thick boundary layer (Figure 4).


**Coronary Angiographic Perspective:**


In coronary arteries, the intima relies on a continuous supply of oxygen and nutrients directly diffused from the circulating blood [18]. When flow along the vessel wall slows, angiography often shows a thickened contrast layer, reflecting a low-shear, stagnant zone. Consequently, oxygen within the boundary layer is replenished more slowly, reducing its availability for direct diffusion to the intima. This sustained hypoxic state may activate hypoxia-inducible factor-1α (HIF-1α), which reprograms intimal cells toward an osteoblast-like phenotype, thereby facilitating calcium deposition [19,20] (Figure 5A–C and Figure 6A,B).

### 2.4. Recirculating Flow


**Fluid Mechanics Perspective:**


When a fluid channel or pipe bifurcates, the distributary experiences a sharper angular deflection compared with the main channel. At the flow divider, central streamlines maintain high velocity, whereas boundary-layer flow along the outer curvature decelerates due to friction. When the velocity gradient between the inner and outer layers surpasses a critical threshold, boundary-layer flow may be entrained into the core and circulate counter to the main flow, forming horizontal vortices or eddies—phenomena collectively defined as recirculating flow (Figure 7A–D) [21].

Within shear zones, velocity gradients generate turbulence and localized erosive forces. In rivers, this manifests as sediment mobilization, particle abrasion, and progressive reshaping of the channel bed and banks. At confluences, interacting currents of disparate momentum and direction produce vortical structures, amplifying turbulence and mixing (Figure 8A,B) [22].


**Coronary Angiographic Perspective:**


In coronary arteries, shear zones occur at sites of flow acceleration, deceleration, or directional deflection, including bifurcations, curvature, and stenotic regions. Disturbed or oscillatory shear stress in these locations compromises endothelial integrity, promoting lipid infiltration and plaque development. High shear gradients at irregular plaque surfaces may destabilize vulnerable lesions, precipitating local rupture and inducing clinical acute coronary syndromes. Over time, chronic exposure to abnormal shear stress drives arterial remodeling, modifying vessel geometry and attenuating the coronary circulation’s capacity to adapt to dynamic flow demands (Figure 9A–I).

### 2.5. Collision Secondary to Water Hammer Shock

In the setting of uncontrolled hypertension, could the abrupt contraction of the LV generate a retrograde pressure wave that propagates at the speed of sound within the arterial wall–fluid system? Is this phenomenon analogous to the water hammer effect observed in pipes and pumps [23]?


**Fluid Mechanics Perspective:**


Consider a tank connected to a pipe that drains fluid. When the distal valve is abruptly closed, the fluid adjacent to the valve stops, whereas the upstream fluid continues to move forward, creating a pressure rise at the point of reflection. The resulting pressure waves travel back and forth within the pipe, where they may reinforce or cancel each another, producing rapid fluctuations in pressure. This transient phenomenon, known as the water hammer effect, imposes substantial stresses on the pipe, induces vibrations, creates noise, and damages the inner lining of pipes and components of pumps [24].


**Coronary Dynamics Perspective:**


In a similar setting of the cardiovascular system, the aorta functions as the tank, and a coronary artery as the draining pipe (Figure 10). During diastole, coronary blood flows forwards. When the LV contracts at the onset of systole—acting as a distal valve—the flow halts at the junction between the epicardial and intramyocardial arteries. The antegrade flow becomes stationary as myocardial pressure rises to match diastolic coronary pressure. Because systolic pressure rapidly exceeds diastolic pressure, LV contraction constricts the myocardial arterioles and capillaries, reverses flow direction, and generates a retrograde pressure wave to collide with the stationary antegrade blood column [25] (Figure 10).


**Acoustics Analysis of In Vitro Studies:**


To analyze the water hammer phenomenon, acoustic principles were utilized to evaluate the retrograde pressure wave propagating at nearly the speed of sound in both pipe systems and coronary arteries. Initially, the review focused on in vitro experiments examining sound-wave propagation through a tubular structure filled with air particles. At baseline, these particles remained undisturbed in their equilibrium positions, uniformly distributed in a random configuration. When a pressure wave traversed the system, the resulting particle motion conformed to fundamental acoustic principles. Instead of driving or compressing the entire column of air, the wave displaced particles only from their equilibrium positions. This displacement manifested either axially—characterized by vertical oscillation at a fixed location—or longitudinally—entailing oscillation along the length of the tube [26] (Figure 11).


**Acoustic Analysis of Coronary Angiographic Flow:**


Within this framework, analogous acoustic phenomena were suspected in coronary angiography, as evidenced by patterns of contrast distribution. Specifically, after the injected contrast filled the target coronary artery, the resulting pressure waves underwent reflection and interference, creating distinct alternating localized regions of remnant contrast concentration emerged during contrast clearance. They were localized high-density contrast zones (compression zones, classified as antinodes), localized moderate-density contrast zones (rarefaction zones, also categorized as antinodes), and segments with minimal contrast retention (nodes) in between (Figure 12A). These acoustic nodes and antinodes concentrate mechanical stress, producing abrupt fluctuations in intraluminal pressure and predisposing the intimal layer to injury. The spatial distribution of these antinodes and nodes, along with their correlation to corresponding lesions, is highlighted in Figure 12B.


**Acoustic Mechanisms of Coronary Injuries:**


Within the framework of coronary artery dynamics, the antegrade blood flow is dominant during diastole. At time zero—marking the onset of systole—a retrograde pressure wave, generated by a water hammer effect, propagates backward from the distal end of the coronary artery. These hemodynamic events reflect the interactions between fluid dynamic processes and acoustic wave propagation (Figure 10).

The particles were concentrated in high density in the zone of compression, alternate with areas of moderate density (rarefaction). The zones of compression with maximal oscillations depicted high pressure and turbulence. The zone of rarefaction with minimal oscillation had lower pressure fluctuation so less turbulence. In the segments between compression and rarefaction, there was no particle movement (Figure 11). A similar pattern of air bubble pockets within compression and rarefaction zones leading to cavitation has been reported in oil pipeline transport, where pressure waves propagated through the system [27]. Furthermore, cultured cells clustered in compression and rarefaction patterns has been documented in culture tubes exposed to pressure waves propagation [28].

At location #1 within the compression zone, the low-pressure antegrade flow during diastole is abruptly displaced by a high-pressure retrograde surge at the onset of systole, generating a pronounced pressure crest. This sudden impulse disrupts the endothelial integrity of the intima, thereby promoting lesion formation, progression and potential rupture (Figure 12B).

In contrast, at location #2, the high-pressure systolic flow diminishes to a lower diastolic pressure. The absence of pronounced pressure surges at this rarefaction site results in minimal turbulence and consequently limited intimal injury. As a result, lesions at this location are typically mild to moderate. Moreover, in coronary arteries, their limited length restricts the buildup of pressure momentum, thereby reducing the likelihood of significant damage (Figure 12B).

At location #3, a low-pressure surge occurs due to the marked decline in pressure at the distal end of the artery. Lesions in this region develop gradually through the progressive deposition of atherogenic material. In the intervening segments between these lesions—whether nodes or antinodes—pressure fluctuations and particle motion remain minimal. This lack of substantial dynamic stress restricts lesion initiation and progression; consequently, these segments exhibit minimal or no lesions (Figure 12B).

### 2.6. Codification of Lesions and Creation of a Coronary Acoustic Map

Leveraging the spatial distribution of these defined segments and sites, an acoustic coronary map can be constructed to systematically codify existing lesions and prospectively track the emergence of new ones (Figure 13) [29]. Building on these mechanistic insights, the following section outlines study protocols that integrate frame-by-frame angiographic analysis with artificial intelligence algorithms to rigorously evaluate the diagnostic performance and the predictive accuracy of the coronary acoustic map across larger populations.

## 3. Study Design and Protocol


**Goals and Challenges:**


The study aims to validate the incidence and temporal patterns of lesions at four predefined sites (#1–#4) on the coronary acoustic map, with particular emphasis on site #1. The objectives include detecting and confirming these lesions through frame-by-frame angiographic analysis and AI-based algorithms. Furthermore, the study seeks to elucidate the mechanisms underlying flow dynamics, pressure surges, and potential intimal injury. Given the challenges of directly measuring flow dynamics in coronary arteries with diameters < 3 mm, it is essential to identify surrogate markers—such as contrast concentration or movement as acoustic signals—to infer flow turbulence and fluctuations in intraluminal pressure.


**Methodology: Theoretical Basis of Contrast as Acoustic Signals:**


When visualized through dynamic coronary angiography, the contrast medium behaves in a manner analogous to acoustic wave propagation. Areas of high contrast density represent compression zones—akin to antinodes in standing wave patterns—while regions with moderate contrast represent rarefaction zones. These wave-driven disturbances may leave subtle yet detectable signatures on high-frame-rate coronary angiography (15 frames per second). Rapid contrast motion and localized variations in contrast density—such as compression (convergence) or rarefaction (divergence) patterns—may serve as visual indicators of transient pressure disruptions and wave interactions (Figure 12A).

By treating contrast propagation as an acoustic signal, angiographic frame to frame analysis was performed and AI models can be trained to recognize patterns in contrast retention and washout that correlate with wave interference or reflection. These temporal and spatial fluctuations in contrast behavior are not random; they follow fluid dynamic laws and can be quantified using metrics derived from pixel intensity and motion over sequential frames. In this way, contrast flow becomes a surrogate for acoustic signal mapping, enabling the detection of functional abnormalities even in the absence of structural stenosis.

The spatial coordinates where these interactions consistently cancel or trap energy can be conceptualized as “acoustic nodes,” analogous to nodal points in physical acoustics. These nodes may represent biomechanically vulnerable regions where flow inertia is reduced, residence time is prolonged, and shear stress becomes irregular or oscillatory—conditions that have been strongly associated with endothelial dysfunction, lipid deposition, and early plaque formation. This new analysis of coronary flow from an acoustic perspective by angiographic protocol and AI algorithm allows the identification of nodal points and high-risk zones before structural abnormalities become radiographically visible. In this sense, coronary flow analysis evolves from a pressure-centric to a wave-centric paradigm, enhancing both diagnostic accuracy and physiologic interpretation.


**Inclusion criteria:**


All patients who underwent coronary angiography for unstable angina. The patients were included in the study if their angiogram showed stenosis in one of the three major coronary arteries. Patients with mild to moderate lesion in the RCA, LAD or LCX arteries were selected.


**Exclusion criteria:**


The patients were excluded if they had chronic total occlusion, previous bypass graft surgery (CABG), or prior percutaneous intervention (PCI) or had been admitted for STEMI.


**New dynamic angiographic recording technique:**


The recording began with contrast injection and continued until complete opacification of the coronary arteries. After the injection ceased, the camera continued capturing images as blood (appearing white) entered and displaced the black-appearing contrast medium. This displacement enabled visualization of flow patterns, their structural features, and temporal duration. Flow could be laminar or disorganized, anterograde or retrograde, and either transient or sustained. The coronary angiogram was acquired at 15 frames per second, corresponding to 0.067 s (67 ms) per frame [3].


**Protocol of angiographic frame to frame analysis:**


In the new dynamic angiographic technique, the aim was to identify accurately the presence or absence of lesions in the four specific locations on the coronary acoustic map. This study also aimed to assess whether they coincided with the location where diastole transitioned to systole or systole to diastole. The lesions classified as location #1 were prioritized because of their vulnerability and elevated rupture risk. These lesions were most frequently observed in the mid-RCA or at the distal end of the proximal segment of the LAD and LCX. All the angiographic results were independently interpreted by two experienced interventional cardiologists, and they were blinded to each other’s assessment. In case of disagreement, the lead author made the final decision.


**Data Collection:**


In this pioneering study on the acoustic map of coronary arteries, to build a comprehensive analytical model, data were collected across several key domains. The foundation was coronary angiography videos with the novel analysis. The location, stenosis degree, wash-out time of contrast were included. Patient demographics and lifestyle factors, including age, sex, body mass index (BMI), and smoking were recorded. Past medical history was thoroughly reviewed, encompassing diabetes, hypertension, dyslipidemia, chronic kidney disease (defined by an estimated glomerular filtration rate <60 mL/min/1.73 m^2^), and a family history of premature cardiovascular disease. Furthermore, laboratory parameters were analyzed, including complete blood count indices (hemoglobin, hematocrit), a full lipid profile (total cholesterol, LDL, high-density lipoprotein (HDL), triglycerides), and biomarkers such as hemoglobin A1c.


**Statistical analysis:**


Continuous variables such as age, body mass index (BMI), the number of lesions, and lipid levels (LDL-cholesterol, HDL-cholesterol, triglycerides, and total cholesterol) were expressed as mean ± standard deviation (SD) if normally distributed and compared using the independent Student’s *t*-test. Non-normally distributed continuous variables were summarized as median with interquartile range (IQR) and compared using the Mann–Whitney U test. Categorical variables—including sex, past medical history (diabetes, hypertension, dyslipidemia), smoking status, and family history of coronary artery disease—were presented as frequencies and percentages. Group comparisons for these categorical variables were performed using the Chi-squared test or Fisher’s exact test, as appropriate (the latter when more than 20% of expected cell counts were less than 5).

Subsequently, univariate logistic regression analyses were conducted to assess the individual association between each potential risk factor and the compression site. Variables that demonstrated a significant association in the univariate analysis (typically with a *p*-value < 0.20) were then included in a multivariate logistic regression model for each position. This final model was adjusted for clinically relevant confounders to identify independent predictors of compression site. All data were stored by excel and analysis with Stata/MP 18 (Statacorp LLC, College Station, TX, USA).


**Artificial Intelligence Algorithm:**


In this protocol, the primary objective is to delineate and quantify the four zones within the coronary arteries exhibiting pockets of high-contrast density (compression) and moderate-contrast density (rarefaction), designated as antinodes (Figure 12A). In contrast, areas with minimal residual contrast are classified as nodes. These classifications are finalized at the end of the second cardiac cycle when the contrast is almost all washed out, leaving pockets of remnant contrast representing the most likely zones of compression or rarefaction. The protocol further aims to correlate these zones with the presence or absence of luminal stenosis or lesions.


**General Workflow:**


The initial step involves segmenting images of the coronary arteries. For this purpose, two models are trained using the R2U-Attention architecture on two distinct datasets: one comprising labeled full-vessel images and the other consisting of labeled catheter images. The first model is trained to detect vessel edges in X-ray images, which may also include catheters misclassified as vessels. The second model is trained specifically to detect catheters, enabling their removal from the output of the first model. This approach ultimately yields segmented images of the coronary vasculature (Figure 14).

Following the acquisition of these segmentation images, which exclude catheters, we will create sliding windows of that slide across the vessels. Each window will capture the coordinates (x, y) and average color value (c) in the window and save them as a matrix for the next steps. For each vessel image, we will use M = 80 windows. If the vessel is small and only uses fewer windows, all other values will be set as 0. If the vessel is large, we will stop sliding after window M since most remaining parts are very small. After this step, for each image, we will have a matrix with a size of M × 3 (x, y, c). This number M is set based on our observation with vessel images since M windows is quite enough to capture important information about the images.


**Dataset Preparation:**


The coronary angiograms were retrieved from the hospital’s Picture Archiving and Communication System (PACS) and screened against the study’s inclusion criteria. The raw Digital Imaging and Communication in Medicine (DICOM) images were converted to the Joint Photographic Expert Group (JPEG) format to streamline model training, as this format offers a favorable balance between file size and sufficient image quality. This conversion significantly reduced storage and computational demands, thereby enhancing data processing efficiency. While minimally compressed, the JPEG images retained the essential anatomical details required for the model to identify subtle angiographic patterns and maintain high classification accuracy.

The angiographic sequences, captured at 15 frames per second, were decomposed into individual grayscale frames. For each frame, corresponding segmentation masks were generated manually by experienced interventional cardiologist. Annotations of compression sites were incorporated to mark lesion location and severity. Furthermore, frames were labeled based on the cardiac phase—diastole, systole, or transitional states. This dataset serves as the foundation for both model training and the subsequent quantitative analysis of contrast behavior over time.


**Data Preprocessing:**


Pre-processing begins by identifying the frame at which contrast opacification is maximal, typically represented by the lowest mean grayscale intensity across the vessel mask. This frame marks the start of the dynamic flow analysis. From this point forward, all subsequent frames are used to assess contrast washout. The pixel values within each frame are normalized, and segmentation masks are applied frame-by-frame to account for vessel movement. For each segmented vessel region, the mean grayscale intensity is computed, particularly in the final frames of the angiogram, to quantify remnant contrast concentration. Regions that retain a high contrast signal are likely to correspond to areas of impaired flow or pressure turbulence. Sliding window averaging may be used to smooth short-term fluctuations across consecutive frames.


**Model Architectures:**


The AI pipeline integrates two main model types. The first is a segmentation network based on a modified Residual Recurrent U-Net (R2U-Net) with attention mechanisms. This architecture is well-suited for capturing fine, dynamically moving vascular structures and provides accurate delineation of coronary vessels across time. The second model is a temporal classification network, implemented as a lightweight Transformer. It receives sequences of segmented frames and learns to predict the likelihood of each region belonging to one of the four lesion-prone sites (1, 2, 3, 4) based on the timing of the frame, anatomical context, and contrast retention.


**Training and Evaluation:**


Both segmentation and classification models are trained in a supervised fashion. The segmentation network is optimized using binary cross-entropy and Dice loss to handle class imbalance and ensure spatial accuracy. The classification network is trained using cross-entropy loss and evaluated using precision, recall, and F1-score to assess its ability to identify lesion locations and severity correctly. For remnant contrast estimation, mean absolute error is used as a regression metric. Evaluation is performed through patient-level cross-validation to prevent data leakage and ensure model generalizability. Together, these methods enable a physiologically grounded, data-driven approach to identifying flow disturbances and predicting lesion formation in dynamic coronary angiography.

AI models trained on temporal angiogram data, through segmentation and intensity analysis, can classify each zone by analyzing these acoustic-equivalent features. This mapping allows the AI system to detect where a lesion is most likely to develop and characterize the mechanism of injury based on wave dynamics, providing both spatial and physiological insight into coronary artery disease progression.

## 4. Conclusions

Grounded in fluid mechanics and acoustic principles, this analysis provides a scientific framework for angiographic evaluation of hemodynamic disturbances which hypothetically undermine endothelial integrity in coronary arteries. The first section examines injuries resulting from repetitive flexion and extension of coronary segments during left ventricular contraction, particularly at the transition from diastole to systole. The second section analyzes the hypothetical impact of thickened boundary layers and oxygen-deprivation-induced intimal injury along the proximal portion of the outer curvature of side branches. The third section addresses the hypothetical contribution of recirculating flow to lesion development at the exit slope of the ostium of side-branch. The fourth section introduces an acoustics-based diagnostic approach for assessing the hypothesis of retrograde pressure-wave propagation linked to water-hammer phenomena. Collectively, these mechanisms define the systematic codification and spatial localization of coronary lesions, as represented on the coronary acoustic map. Building on these insights, the present analysis outlines a clinical trial framework incorporating AI-based algorithmic protocols to rigorously evaluate the diagnostic performance and predictive accuracy of the coronary acoustic map.


**Limitations:**


This study has several limitations that should be acknowledged. First, the analysis was conducted under simplified angiographic and hemodynamic assumptions that may not fully capture the complex mechanical environment of coronary arteries in vivo. Although images of coronary angiogram provide valuable insights, they inevitably omit certain dynamic parameters such as vessel wall elasticity, myocardial motion, neurohumoral regulation or calcified versus soft plaque cover. Second, the evidence presented is retrospective and observational, based on selected angiographic cases, and thus insufficient to confirm causality. Methodological concerns also arise from the reliance on contrast dynamics as a surrogate for acoustic signals, which have not been validated against invasive hemodynamic standards such as pressure-wire or Doppler flow. Third, the imaging resolution and temporal sampling rate (15 images per second) constrained the accuracy of flow field reconstruction, particularly during rapid changes in systolic–diastolic phases. Fourth, the study primarily focused on idealized boundary conditions and steady or quasi-steady flow regimes, whereas real coronary flow exhibits strong pulsatility and transient phenomena, including pressure wave reflection and flow reversal at different level of heart rates and pressures.


**Future Plan:**


The future plan includes using open-source software or commercial platforms to incorporate prospective clinical trial and AI-based algorithmic protocols to rigorously evaluate the diagnostic performance and predictive accuracy of the coronary acoustic map.

## 5. Perspective

**What Is Known?** Atherosclerosis begins with injury to the endothelial cells, allowing LDL cholesterol particles to cross the intima, accumulate at the sub-intimal space and gradually aggregate into plaques.

**What Is New?** Leveraging the spatial distribution of the lesions, an acoustic coronary map could be constructed to systematically codify existing lesions and prospectively predict future ones.

**What Is Next?** Building on these insights, the future plan includes using open-source software or commercial platforms to incorporate prospective clinical trial and AI-based algorithmic protocols to rigorously evaluate the diagnostic performance and predictive accuracy of the coronary acoustic map.

## Figures and Tables

**Figure 1 diagnostics-15-02994-f001:**
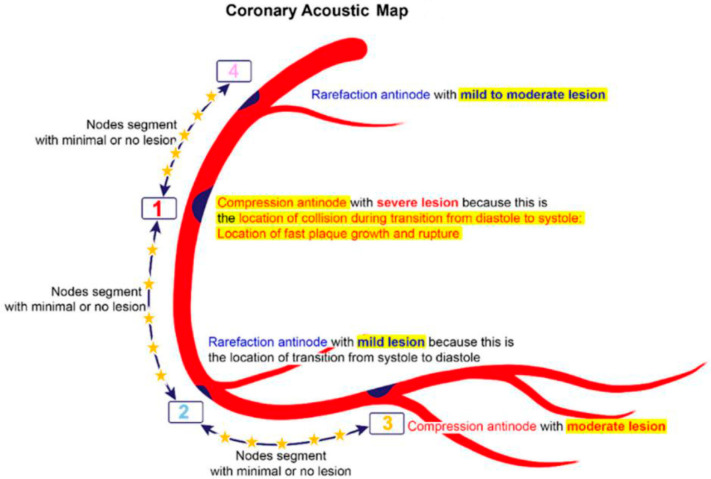
**Coronary acoustic map.** In this map, the moderate to severe lesions are at the compression zones while the mild lesions are at the rarefaction zone. The lesions from location #4 to location #1 happen mainly by disturbances from an antegrade direction during diastole while the lesions from location #1 to location #2 happen mainly by disturbances from retrograde direction during systole. This is the first time we classify lesions based on temporal criteria. Reproduced from reference [12].

**Figure 2 diagnostics-15-02994-f002:**
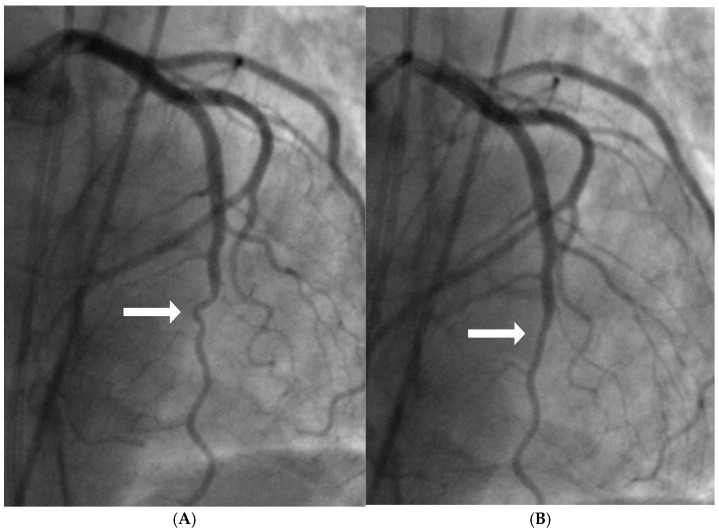
(**A,B**) Repetitive flexion and extension at hinge location. At the middle of the artery, the segment bends with systole and straightens during diastole (arrow).

**Figure 3 diagnostics-15-02994-f003:**
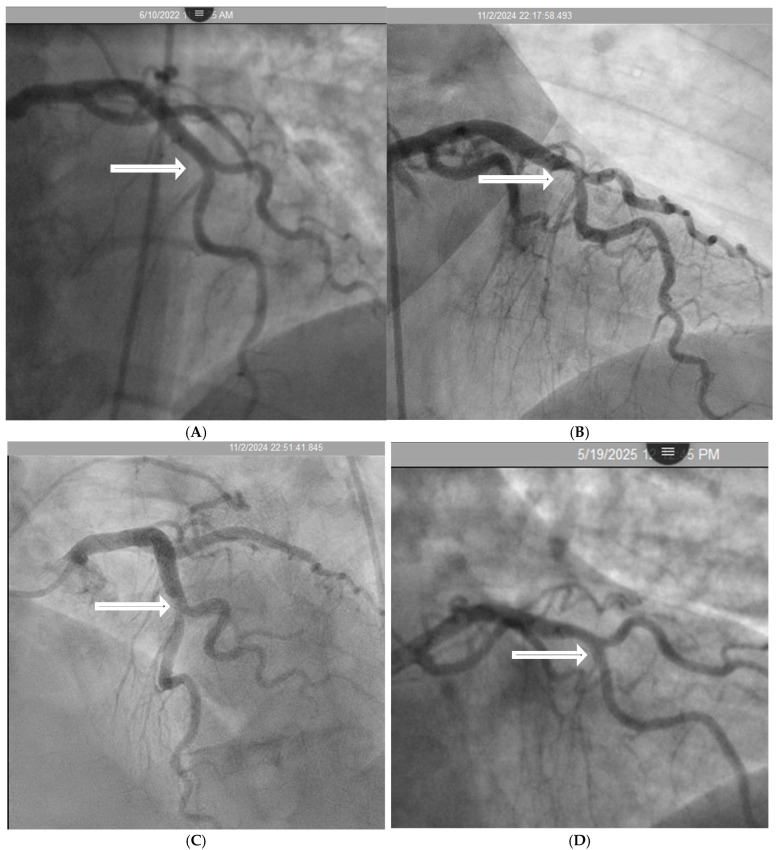
(**A**–**D**) Changes in severity of lesion of the non-infarct related artery. (**A**). This is the left anterior descending artery (LAD) in 2022 without lesion at its midsegment after the origin of a large diagonal (arrow). (**B**). In 11/2024, after acute occlusion of the large right coronary artery (RCA) there was compensatory hyperdynamic contraction of the left ventricle (LV) with increased haziness at the midsegment of the LAD (arrow). (**C**) After opening the RCA which was the infarct-related artery, another angiogram of the LAD showed mild to moderate stenosis which was not as bad as in (**B**) (arrow). (**D**) Seven months later in 5/2025, the coronary angiography showed no lesion at the mid LAD (arrow). The exact date and time of the angiographic images are specified at the top of the figure.

**Figure 4 diagnostics-15-02994-f004:**
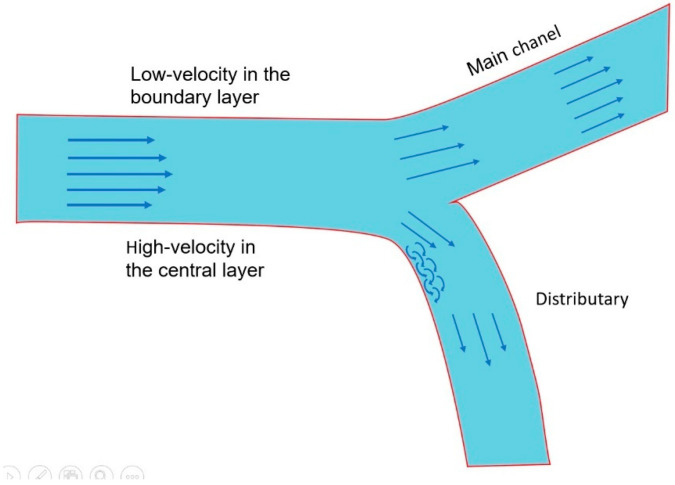
Reorganization of flows in main channel and distributary with development of thick boundary layer at the outer curve of the distributary.

**Figure 5 diagnostics-15-02994-f005:**
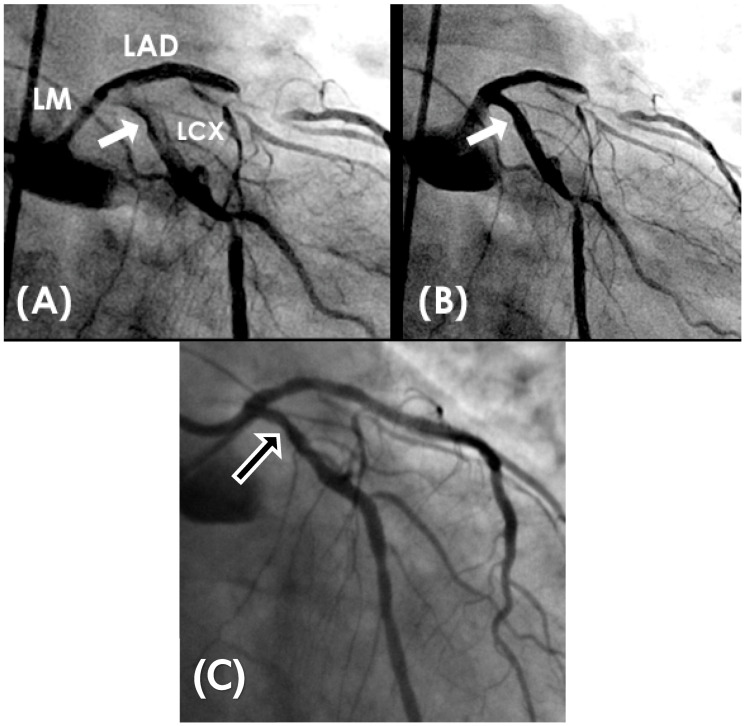
(**A**–**C**) Thick boundary layer in 2011and 2019. (**A**) In 2011, the coronary angiogram showed a thick boundary layer (white arrow) at the proximal segment of the left circumflex (LCX) (arrow), LAD= left anterior descending artery, LM= left main coronary artery. (**B**) Here there was minimal narrowing (arrow). (**C**). Eight years later in 2019, patient had stenting of the lesion at the mid LCX. The severity of the proximal LCX had no change (arrow).

**Figure 6 diagnostics-15-02994-f006:**
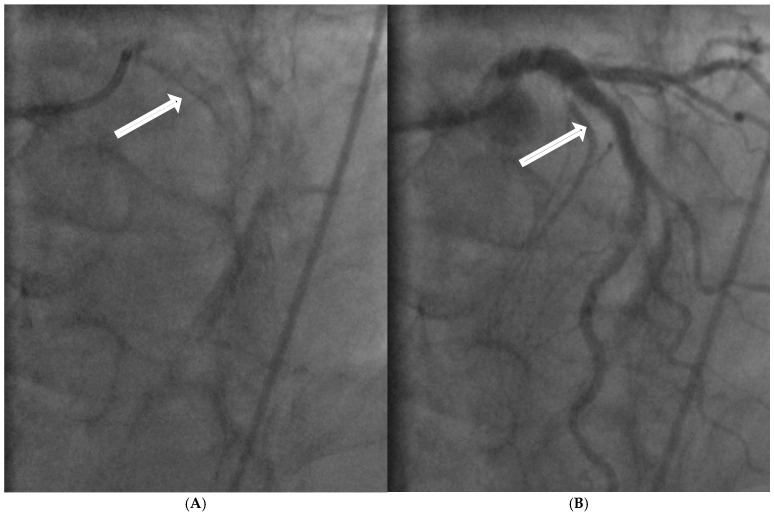
(**A**,**B**) Calcified segment without significant stenosis. (**A**) The proximal segment of the left anterior descending (LAD) artery showed calcification (arrow) while there was none in other segment. (**B**) This segment had minimal stenosis (arrow). Reproduced from reference [12].

**Figure 7 diagnostics-15-02994-f007:**
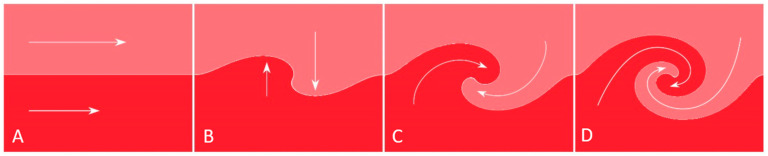
(**A**–**D**). Recirculating Flow. (**A**) Initially, the central layers exhibit higher velocities, while the peripheral layers slow down due to friction at the wall. (**B**) When the velocity differential reaches a critical threshold, (**C**) the innermost portion of the peripheral layers becomes entrained toward the center (**D**), generating recirculating flow. Reproduced from reference [12].

**Figure 8 diagnostics-15-02994-f008:**
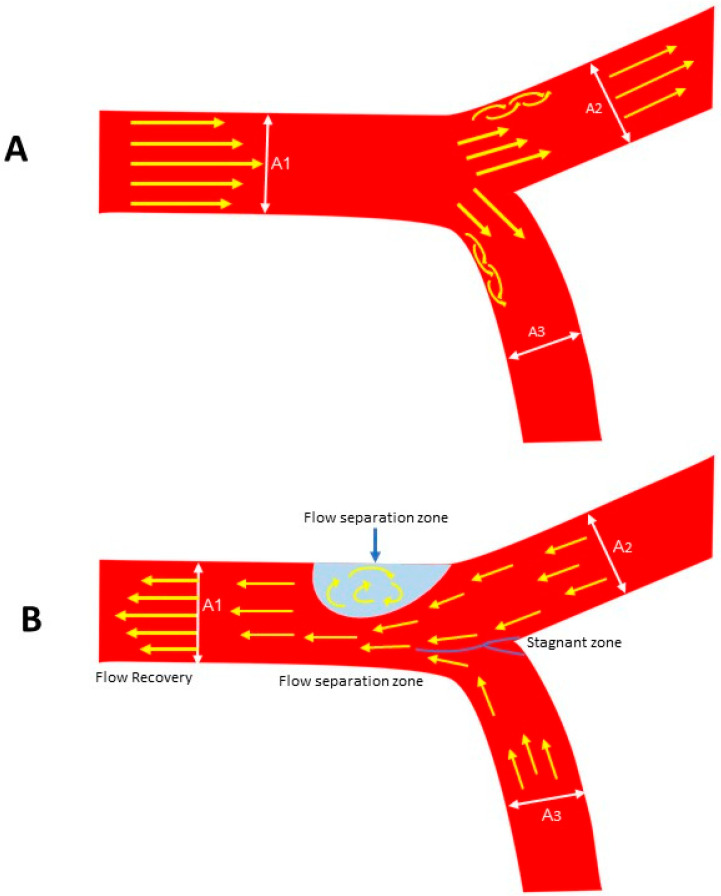
(**A**,**B**) Lesions proximal to the bifurcation or distal to the divergent point. (**A**) In this model, the flow is antegrade. The flow recirculates at the outer curve of the side branch. (**B**). Here the lesion is proximal to the bifurcation because the flow is retrograde and the recirculating flow is at the outer curve of the large main vessel. A1: Main vessel; A2: Main branch; A3: Side branch.

**Figure 9 diagnostics-15-02994-f009:**
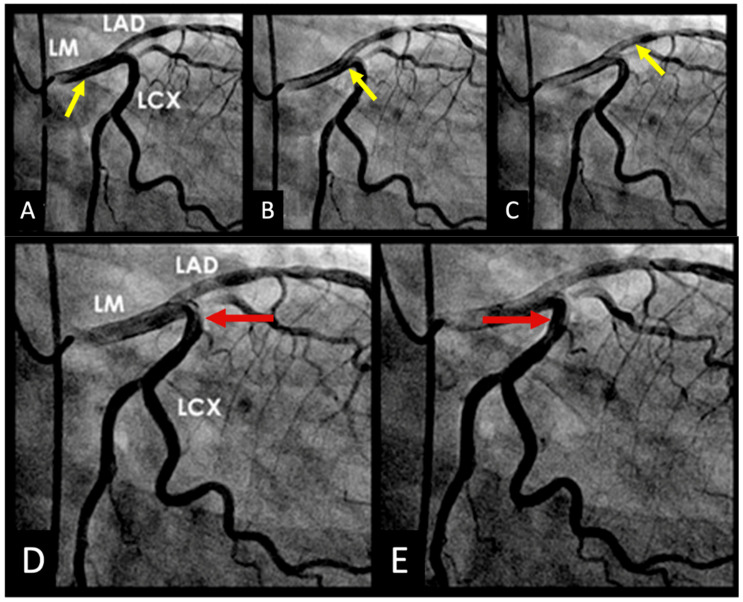
(**A**–**I**) Coronary Recirculating Flow at the Outer Curve of a Side Branch This is a series of nine consecutive coronary angiographic images, filmed at 15 images per second or the interval between consecutive images = 0.067 s or 67 ms. (**A**) At the beginning of diastole, the blood in white color enters the left main (LM) (yellow arrow). (**B**) 67 ms later, in the next image, the blood in white moves to the proximal left anterior descending artery (LAD) (yellow arrow) without yet entering the left circumflex artery (LCX). (**C**) 67 ms later, the blood moves down the mid LAD (yellow arrow) while it is still at the ostium of the LCX. (**D**) In a higher magnification, the flow is at the ostium of the LCX, preferentially turns at the curve, along the carina (red arrow). This is the end of diastole. (**E**) 67 ms later, at the beginning of systole, the blood is at the proximal and mid LAD, while at the LCX, the blood just barely enters the LCX on the carina side. There is a thick layer of contrast at the outer curve of the LCX, opposite to the carina (red arrow). (**F**) At this stage, the blood moves more slowly at the mid LAD (yellow arrow (**F**–**I**)), because this is the 2nd image of systole while at the LCX, the blood is still at the ostial and proximal segment. The peripheral layer is still prominent with some elements of white blood (red arrow). This is the evidence of recirculating or swirling flow at the boundary layer. (**G**) The flow on the carina border is still homogenously white, even the speed is slow, because this is during systole. At the same time, there is turbulent movement of the blood (mixing of black and white) inside the boundary layer (red arrow). (**H**) At the LCX, there is more turbulence at the peripheral layer (red arrow). (**I**) At this stage, the antegrade flow in the proximal LCX is larger, more prominent, and occupies more than half of the proximal LCX. The boundary layer at the proximal LCX is now limited to a thin layer. The black color of the boundary layer is more homogenous. It may be due to reversed flow during systole (red arrow).

**Figure 10 diagnostics-15-02994-f010:**
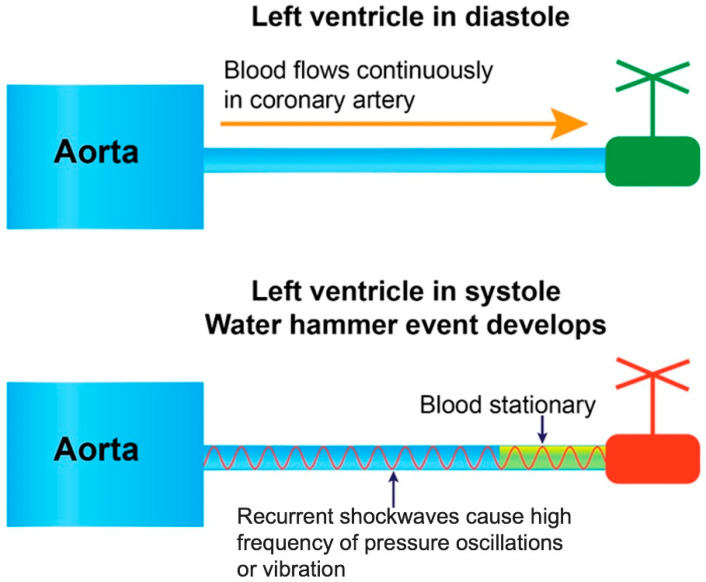
Water Hammer in Coronary Artery. A water-hammer phenomenon may arise in the coronary arteries when a sudden change in flow velocity—induced by abrupt left ventricular contraction—generates a retrograde pressure wave propagating at approximately the speed of sound. Reproduced from reference [12].

**Figure 11 diagnostics-15-02994-f011:**
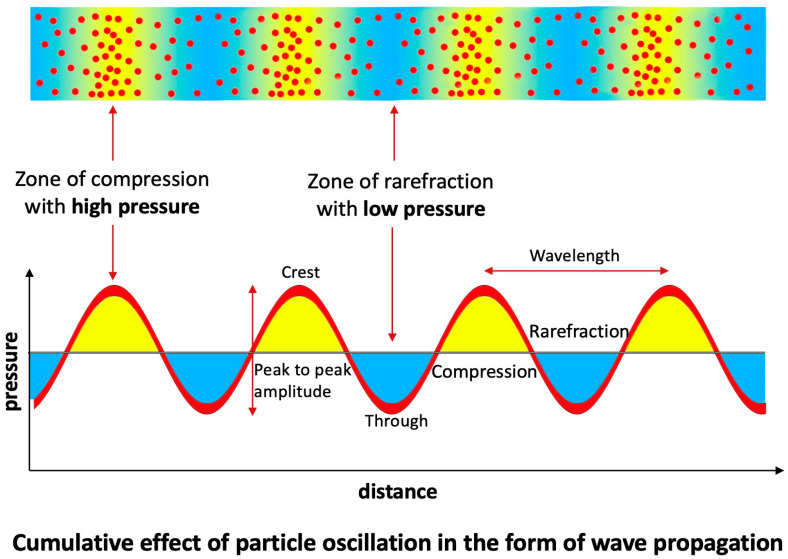
In a tubular experimental setup, the passage of a pressure wave induced particle motion consistent with fundamental acoustic principles. Air particles aggregated at high density within compression zones, alternating with regions of reduced density (rarefaction). Compression zones exhibited maximal oscillatory amplitude associated with elevated pressure and increased turbulence, whereas rarefaction zones displayed minimal oscillation, characterized by lower pressure fluctuations and correspondingly reduced turbulence.

**Figure 12 diagnostics-15-02994-f012:**
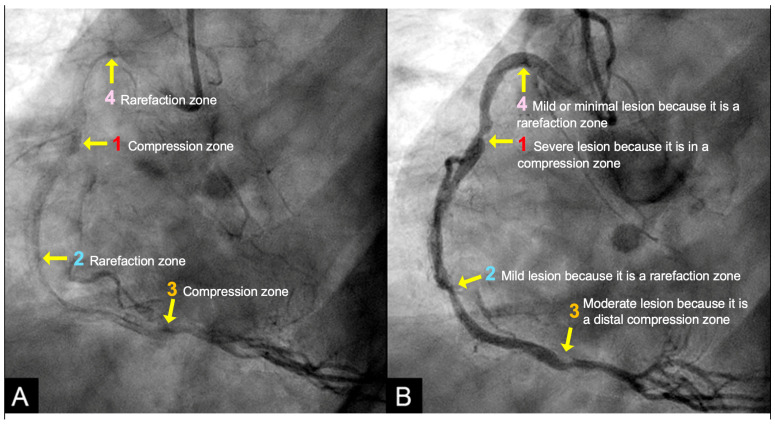
(**A**,**B**) This angiogram of the right coronary artery, captured at end-diastole when most of the contrast has been cleared, shows residual pockets of high contrast concentration. These stagnation zones denote regions of elevated or moderate contrast density and correspond to areas of compression and rarefaction at the sites of antinodes. By contrast, the intervening coronary segments, corresponding to nodes, appear free of lesions, potentially because they are not subjected to direct interaction with the propagating pressure wave. Reproduced from reference [12].

**Figure 13 diagnostics-15-02994-f013:**
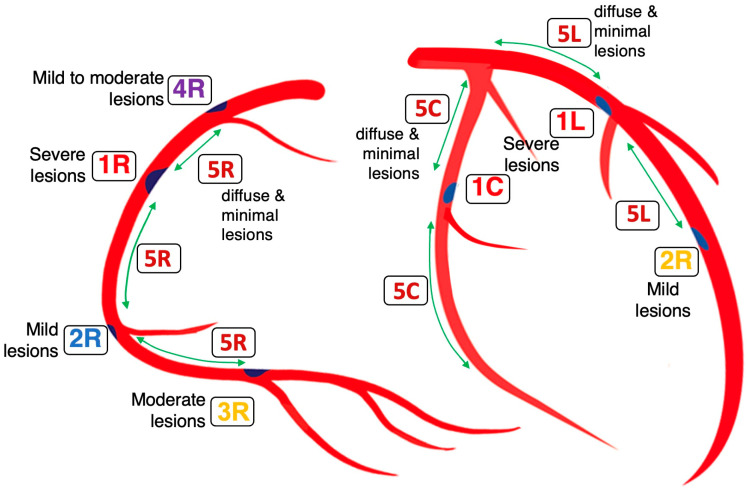
**Coronary Acoustic Map.** In the coronary arteries, the most severe lesion happens at the compression antinodes (location #1) or moderate lesion at #3, while mild lesion happens at the rarefaction antinodes (#2 and #4). Conversely, the segments in between the antinodes 5R, 5C and 5L have diffused minimal or no lesion due to minimal pressure fluctuations. These nodal sites consistently occur at the dynamic inflection points of the cardiac cycle: the diastole-to-systole transition (locations #1 and #3) and the systole-to-diastole transition (location #2). Because the diastole-to-systole transition involves a sharp escalation of pressure and constructive interference of forward and backward waves, nodal stresses at location #1 are particularly intense, resulting in more aggressive lesion growth. At location #2, the de-escalating pressure wave during systole-to-diastole transition reduces stress intensity, leading to slower lesion progression. Lesions at location #3 also develop gradually; here, lower systolic pressure diminishes nodal stress, and plaque accumulation is driven predominantly by LDL cholesterol deposition rather than wave-induced mechanical damage. (R = right coronary artery, C = left circumflex artery, L = left anterior descending artery.)

**Figure 14 diagnostics-15-02994-f014:**
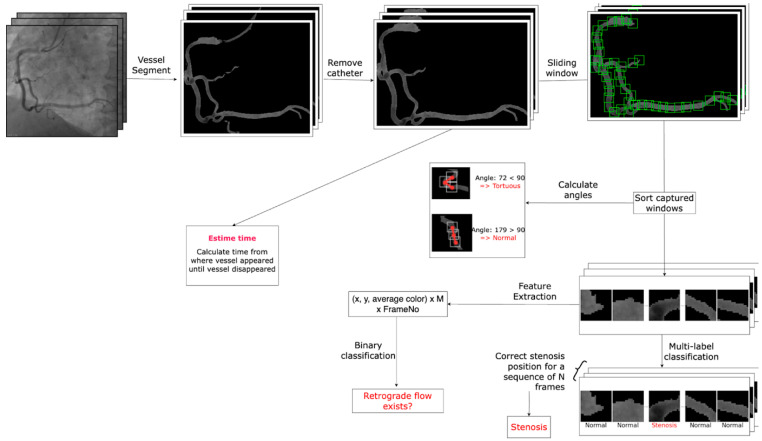
General workflow of the Artificial Intelligence Algorithm Protocols.

**Table 1 diagnostics-15-02994-t001:** **Mechanisms of coronary atherosclerosis**.

1.	Mechanical stress due to repetitive flexion and extension at a hinge location
2.	Local intimal ischemia due to thick boundary layers
3.	Recirculating flow at an exit curve of a bifurcation
4.	Excessive pressure fluctuation at compression nodes generated by collision with a retrograde pressure wave

## Data Availability

The original contributions presented in the study are included in the article. Further inquiries can be directed to the corresponding author.

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
