# Peer review of "Codify and Localize Lesions on a Coronary Acoustic Map: Scientific Rationale, Trial Design and Artificial Intelligence Algorithm Protocols"

_diagnostics, 2025, doi:10.3390/diagnostics15232994_

Round 1
Reviewer 1 Report
Comments and Suggestions for Authors
Dear authors!
The model you presented for constructing an acoustic map of the coronary artery is innovative and has the potential to change existing approaches to coronary artery research.
Suggestions for improving the manuscript:
- In the introduction, please describe in more detail existing biometric models of atherogenesis and how this model relates to these paradigms.
- Include in the Discussion an analysis of fundamental works about vascular biomechanics and studies that observed similar wave-induced phenomena in biological systems.
- Describe in more detail the limitations of the study.
- Please, check using of abbreviations (some were entered twice, some were not entered at all in the text, eg. LDL on lines 50 and 88, LAD on lines 139-140 without full variant, LV on lines 86 and 127 and 248, STEMI on 126 and 407, etc.)
- Standardize the design of the sources ( MA, Hashim MJ, Mustafa H, Baniyas MY, Al Suwaidi SKBM, AlKatheeri R, Alblooshi FMK, Almatrooshi MEAH, Alzaabi MEH,
Al Darmaki RS, Lootah SNAH. Global Epidemiology of Ischemic Heart Disease: Results from the Global Burden of Disease Study. Cureus. 2020 Jul 23;12(7):e9349. doi: 10.7759/cureus.9349. PMID: 32742886; PMCID: PMC7384703. and 3. Nguyen, T.; Ngo, K.; Vu, T.L.; Nguyen, H.Q.; Pham, D.H.; Kodenchery, M.; Zuin, M.; Rigatelli, G.; Nanjundappa, A.; Gibson, M. Introducing a Novel Innovative Technique for the Recording and Interpretation of Dynamic Coronary Angiography. Diagnostics 2024, 14, 1282. https://doi.org/10.3390/diagnostics14121282.).
Author Response
Please see

Reviewer 2 Report
Comments and Suggestions for Authors
The manuscript entitled “Codify and Localize Lesions on a Coronary Acoustic Map: Scientific Rationale, Trial Design and Artificial Intelligence Algorithm Protocols” by Nguyen et al. presents an innovative framework that integrates fluid mechanics, acoustic principles, and artificial intelligence algorithms to codify and localize coronary lesions using a coronary acoustic map (CAM). The authors propose four mechanistic pathways for lesion development and illustrate these concepts with angiographic examples.
One of the major strengths of this work lies in its novelty and clinical significance. The concept of moving from static morphology-based assessment of stenosis to dynamic evaluation of flow disturbances is both original and highly relevant.
Despite these strengths, the manuscript also has important limitations. The evidence presented is retrospective and observational, based on selected angiographic cases, and thus insufficient to confirm causality. The proposed trial design is ambitious but lacks key details, such as sample size justification, clinical endpoints, and feasibility considerations, particularly in terms of reproducibility and operator independence. Methodological concerns also arise from the reliance on contrast dynamics as a surrogate for acoustic signals, which is not validated against invasive hemodynamic standards such as pressure-wire or Doppler flow. The statistical analysis plan described is basic and does not match the complexity of the proposed AI models. More detail is needed regarding multivariable approaches and machine learning validation metrics.
There is also a risk of overinterpretation. While the mechanistic pathways are plausible and well-illustrated, they should be framed as hypotheses rather than definitive causal mechanisms. The language in several sections could be improved. A glossary or simplified terminology would help to improve accessibility. Finally, while the references are appropriate, greater integration of existing coronary biomechanics and endothelial biology literature could be included.
In summary, this manuscript is highly original and potentially impactful. My overall recommendation is major revision.
Comments on the Quality of English LanguageCould be improved
Author Response
Dear Sirs and Madams
Our team (Khiem D Ngo MD Department of Geriatrics, University of Texas A&M School of Medicine, Temple TX, USA and Thach Nguyen MD, Methodist Hospital, Merrillville IN and St Mary Medical Center, Hobart IN, and C Michael Gibson, Baim Institute of Clinical Research, Harvard Medical School, Boston) would like to THANK YOU for your IMPORTANT SUGGESTIONS.. We re-write and edit many paragraphs based on their recommendations
1. We re-write the statistical analysis with multivariable approaches and machine learning validation metrics.
2. While the mechanistic pathways are plausible and well-illustrated, we frame the explanations as hypotheses
With my warmest personal regards
Thach Nguyen MD FACC FSCAI
